# Bacterial surface interactions with organic colloidal particles: Nanoscale hotspots of organic matter in the ocean

**Nirav Patel**[1,2]*, **Ryan Guillemette**[1], **Ratnesh Lal**[2], **Farooq Azam**[1]

**1** Scripps Institution of Oceanography, University of California San Diego, La Jolla, California, United States of America, **2** Department of Bioengineering, University of California San Diego, La Jolla, California, United States of America

* niravpatel215@gmail.com

**Data Availability Statement:** All relevant data are within the article and its Supporting Information files.

**Funding:** This study was supported by the Gordon and Betty Moore Foundation (https://www.moore.

## Abstract

Colloidal particles constitute a substantial fraction of organic matter in the global ocean and an abundant component of the organic matter interacting with bacterial surfaces. Using *E. coli* ribosomes as model colloidal particles, we applied high-resolution atomic force microscopy to probe bacterial surface interactions with organic colloids to investigate particle attachment and relevant surface features. We observed the formation of ribosome films associating with marine bacteria isolates and natural seawater assemblages, and that bacteria readily utilized the added ribosomes as growth substrate. In exposure experiments ribosomes directly attached onto bacterial surfaces as 40–200 nm clusters and patches of individual particles. We found that certain bacterial cells expressed surface corrugations that range from 50–100 nm in size, and 20 nm deep. Furthermore, our AFM studies revealed surface pits in select bacteria that range between 50–300 nm in width, and 10–50 nm in depth. Our findings suggest novel adaptive strategies of pelagic marine bacteria for colloid capture and utilization as nutrients, as well as storage as nanoscale hotspots of DOM.

## Introduction

Organic colloids in surface waters of the global ocean play a significant role in the biogeochemical cycling of organic matter and for shaping the microenvironment for individual microorganisms [1–3]. The abundance of organic colloids in marine surface waters ($10^8$–$10^9$ particles $mL^{-1}$; < 120 nm-sized) is two to three orders of magnitude higher than the abundance of free-living bacteria and colloid concentration decreases from the surface to deeper waters [4–7]. Colloidal particles influence the availability and distribution of organic matter through their physical interactions with dissolved organic matter (DOM), influencing the aggregation and dissolution processes of larger organic particles [8–10]. Their interactions with biotic and abiotic surfaces contributes to biofouling and modifying surface sites for increased likelihood of bacterial cell attachment and colonization [11]. Bacteria are exposed to frequent interactions with more abundant colloids within their marine microenvironment. Adsorption and specific attachment of colloids can influence surface nutrient and DOM bioavailability, surface protein

org/), Marine Microbial Initiative (grant 4827) to
FA. The funders had no role in study design, data
collection and analysis, decision to publish, or
preparation of the manuscript.

**Competing interests:** The authors have declared
that no competing interests exist.

activity, and enzymatic degradation [12–15]. Interactions of organic colloidal particles can influence the rates of microbial DOM production, dissolution, and remineralization on a microspatial scale [3, 16–18].

Phytoplankton blooms and cell lysis events are significant sources of colloidal DOM (cDOM), where released labile colloids are available for enzymatic breakdown and bacterial uptake. Nascent organic colloids are also released into seawater from various biological sources as polymeric exudates and as cellular detritus comprised dominantly of proteins, polysaccharides, lipids and nucleic acids [19]. Within a dynamic microenvironment with changing environmental factors, colloidal organic particles present readily available labile DOM for marine bacterial utilization, that are susceptible to extrinsic physical and chemical interactions. Individual colloidal particles can effectively behave as discrete packets of concentrated organic nutrients for a period before degradation, potentially functional as individual nanoscale hotspots. Influxes of viral-induced colloidal DOM can promote aggregation and influence organic carbon transport. The utilization of colloids by bacteria and heterotrophic flagellates can also support biodiversity and community shifts in bacterial populations [20–22].

Bacterial strategies targeting specific labile biopolymers can offer specific species competitive advantages within transient nutrient hotspots like microalgal blooms [23, 24]. Marine heterotrophic bacteria can derive significant physiological benefit from such strategies and chemotactic motility to exploit increased collision encounters with organic colloidal particles within enriched microenvironments, such as phycospheres [25, 26]. Motile bacteria can dominate labile DOM consumption during episodic algal bloom release of organic matter, where interactions with organic colloids potentially offer some fraction of cells a competitive advantage [25]. Bacteria behavior can be modelled as patchy colloids where cell attachment onto bacterial surfaces is mediated by localized adhesive patches; such sites for binding and unbinding surfaces can be opportune sites of selective organic colloid attachment and aggregation on bacterial surfaces [27]. Understanding bacteria-particle interactions can offer insight into the mechanisms underlying bacterial utilization of colloidal organic particles and potential bacterial adaptive strategies that yield competitive advantage over other community members.

Ribosomes are ubiquitous cellular components that are released into seawater along with extracellular release of DOM and phytoplankton detritus. Bacterial ribosomes are molecular assemblies of peptides with ribonucleic acids that are the sites for mRNA transcript translation and protein synthesis [28]. They are comprised of 2 discrete subunits, a large 50S and a small 30S subunit, forming a 21 nm-sized 70S bacterial ribosome particle [29]. After their extracellular release, individual ribosome particles are ubiquitous point sources of labile DOM, presenting as potential discrete nanoscale influxes of organic carbon and nitrogen from bacterial scavenging. As such, ribosomes are ideal model organic colloidal particles for investigating the mechanisms and strategies in marine bacterial interactions with specific organic colloids. Extracellular production of ribosomes occurs across multiple taxa of marine bacteria due to viral lysis, and upon release, ribosomes are stable in seawater for sequencing and taxonomic profiling [30]. Paucity of quantitative data currently limits our understanding of the influence of bacterial on colloidal organic particles, their interaction and degradation rates. Emerging methods in nanoscale imaging are expanding the capability for observing and understanding how marine microbes interact with organic colloids within their microenvironment [31]. Atomic force microscopy (AFM) is a high-resolution microscopy technique that measures the nanoscale topography of surfaces using the raster scanning of a sharpened probe tip under feedback control to maintain a constant physical force. Within microbial ecology, AFM imaging has been primarily used to study the sizes and shapes of various components of the marine microenvironments and their interactions with nanometer resolution. It has been applied for observing finer structures and taking quantitative measurements of microbial biovolumes

within natural assemblages [32–34]. Additional marine applications for AFM have been in the characterization of marine gels and diatom-derived extracellular polymers and the investigation of marine viruses collected from California coastal waters [35–38]. The high-resolution capability of AFM imaging has also been utilized for observing and characterizing organic colloids and biopolymers from natural waters [39, 40]. Expanding upon the finer details of individual components of marine microenvironments, AFM imaging has been used to observe morphological details in host-phage interactions for marine bacterium *Roseobacter denitrificans* OCh114 and for phytoplankton *Phaeocystis globosa* [41, 42]. Malfatti and Azam (2009) using AFM showed potential symbiotic interactions between heterotrophic marine bacteria and *Synechococcus* cells, which has implications in bacterial networks and associations within seawater [33]. Seo et al. used AFM to observe marine bacterial capture of submicron particles and the relative frequency within natural bacterial populations, which has implications for the degradation of colloidal organic particles [43].

The capture of colloidal particles on bacterial surfaces is a potentially advantageous strategy in fluctuating spatiotemporal distributions of colloidal DOM for scavenging heterotrophic microbes. Localization of organic colloids upon bacterial surfaces suggests coupling between capture and uptake of cDOM. In the present study we used E. coli ribosomes as model organic colloids to observe the surface attachment and fate of such particles in seawater microcosm experiments.

## Results and discussion

### Ribosomes as particle films on bacterial surfaces

Extracellular ribosomes in ambient seawater, from 16S rRNA proxy measurements, are present at approximately $10^7$ ribosomes mL$^{-1}$, and can reach elevated concentrations up to $10^9$ ribosomes mL$^{-1}$ in regions enriched after cell lysis [30, 44]. As evidence of direct contact-dependent bacterial interactions with ribosome, fluorescently labelled ribosomes or ribosome fragments added at $5 \times 10^9$ to $5 \times 10^{11}$ particles mL$^{-1}$ were found to cover surfaces of cultured *Alteromonas* sp. ALTSIO within 30 sec (S1A and S1B Fig). Fluorescence micrographs showed varying levels of SYBR Green II-labelled ribosomes coverage on FM 4-64-labelled cells. Most cells presented significant coverage of fluorescence signal indicating ribosome interactions with bacteria surface. A subset of cells showed ribosome fluorescence localized on bacterial edges in addition to overall cell surface.

AFM imaging of ATLSIO and natural assemblages in 0.6-μm seawater filtrate amended with ribosomes showed bacteria surrounded or covered with films of particles (S1C and S1D Fig). Particle features of individual ribosome associated with bacterial cells as contiguous films, forming a zone of enrichment. Particle films on and around cells vary in appearance for cells from 0.6-μm seawater filtrate. In comparison, ALTSIO cells amendment with ribosomes and AFM-imaged immediately showed particle films that cover regions extending up to 200 nm from the cell boundary.

Ribosome depletion was observed in atomic force micrographs of ALTSIO and 0.6-μm seawater filtrate cells deposited onto 0.22-μm polycarbonate filters immediately after ribosome amendment (< 5 min) and after 24 hr incubation (S2 Fig). The significant decrease of particles indicated the depletion of suspended ribosomes after incubation. For ALTSIO after 24 h the presence of particle films is completely diminished with little to no background ribosomes, likely due to bacterial degradation of ribosomes. In contrast, incubation of ribosomes with cells in 0.6-μm seawater filtrate for 24 h had more background ribosomes present, and devoid of any cell-associated particle films surrounding cells, suggesting a lower degree of ribosome utilization compared to ALTSIO cells. Near complete depletion of background ribosomes, in

combination with the observed increase in cell abundance for marine bacterial samples is consistent with utilization and degradation of ribosomes via physical interactions.

Cell-associated particle films were primarily observed as particle-enriched zones, effectively as a halo of colloidal ribosomes, with a higher local concentration of cDOM. The physical characterization of such enrichment zones can be challenging due to physical artefact during sample preparation, including forced deposition upon the filtration surface. Such manipulations flatten and obscure the origination 3-dimensional ultrastructure of the particle-enriched zone onto observation surface plane. This effect may present as particle accumulation around cell periphery of individual bacteria, which can be influenced by filtration artefacts due to calls covering filter pores. These observations support the hypothesis that bacteria can capture organic particles or develop a film-like layer of particles to be later utilized as a substrate repository. AFM micrographs of cells from whole seawater filtrate show observed cells with similar coverage and contiguous films of cell-associated ribosome particles (S1E Fig). The surfaces of select cells were observed with patches of ribosome particles, suggesting intimate attachment and association between some bacterial cells and organic colloidal particles within a natural assemblage in cDOM-enriched marine microenvironments.

The addition of ribosome particles results in the coating of bacterial surfaces with a film of contiguous particles, effectively altering their surface properties and surface-mediated interactions with the marine microenvironment. One manifestation of the surface film is as a zone of enrichment, with particles densely populating the proximal areas around cell edges, as observed in filtered samples. Such zones are suggestive of 3D structure to the particle film in a weakly bound association with the surface and susceptible to disruption from physical forces, such as from filtration or centrifugation. The formation of the surface particle film likely occurs through a quick process of particles populating the bacterial surface through rapid adsorption and attachment (S1 Fig). Variations in the adsorption process can influence bacterial sequestration of organic colloids and in turn influence bacterial behaviors in particle-enriched microenvironments. Bacterial utilization of various types of organic colloidal particles raises the possibility of potential bacterial adaptive strategies in selective transformation of particles and DOM distribution on a submicron scale in the marine microenvironment. Different bacterial species can have varying potential for interaction and particle degradation rates and exert influence on the bioavailability of colloidal DOM. Bacteria have access to colloids smaller than 200 nm due to diffusion and convective transport to the surface. Furthermore, they need to expend energy for larger colloids up to 2 μm to intercept particles through increased collision frequency of particles [11]. In relation to these concentration-dependent processes, bacterial surface capture and retention of colloids is an effective strategy for bacteria to generate nanoscale hotspots of DOM. This results in small, localized regions with high effective concentrations of ribosomal particles. The mechanisms of particle attachment and utilization are not known, nor is the cause for heterogeneity within a neighborhood of cells. Further investigation is needed to determine mechanisms that may contribute to specific particle interaction rate for different bacterial species and the variability within a local population of cells. Additionally, it remains to be determined how the age of colloidal DOM and potential refractory DOM molecules may influence the extent and rates of particle attachment to bacterial surfaces.

Our findings of films of organic colloidal particles on bacterial surfaces have significant implications in the ecophysiology of heterotrophic bacterioplankton, considering that cells can transport utilizable labile DOM on their surfaces. When exposed to the high colloidal concentration, bacterial cells are enrobed within a shroud of particles traveling with a nutrient hotspot, extending the residence time of ambient organic colloids. The surface capture of ribosomes is a quick process that occurs within seconds to minutes, suggesting that with

sufficient concentration, the bacterial surface can become saturated, along with surface protein activities in particle breakdown and nutrient uptake. Brownian motion and convective diffusion of smaller colloids can consistently replenish surface particle films with organic colloids from the ambient microenvironment in elevated particle concentrations.

The apparent bacterial consumption of ribosome particles suggests general utilization of organic particles of similar size as a labile organic source of carbon and nitrogen for bacterial production. The breakdown of colloidal DOM can generate smaller colloidal particles and dissolved organic molecules that readily diffuse, which imposes a limitation on the advantage and utility of organic colloidal particle breakdown for individual bacteria. Any fragmented particles and organic molecules that remain suspended and unattached may be lost to bacteria due to the transience of the interaction. Bacterial surfaces can functionally generate nanoscale hotspots by forming contiguous films of particles due to attachment of ambient colloidal organic particles at elevated concentrations. This transformation of organic matter has implications for the residential times of nascent DOM in the surface waters and its vertical transport in the water column. It is most likely that different bacterial taxa have varying levels of influence on organic colloids, where select members in a bacterial assemblage can drive greater levels of interactions and growth in response to influxes of organic colloidal particles. Further investigation is needed to quantify the variability of such utilization and the bacterial respiration and growth efficiency rates with respect to ambient concentrations of specific organic colloids, and the currently unknown factors that influence these rates.

The effect of being covered with ribosomal particles or associated with a contiguous particle film have implication for individual bacterial activity. The attached particles could effectively alter the properties of bacterial surfaces presented to the external environment and the biochemical activity of the surface proteins, enzymes, and transporters. The presence of cells covered with particle films or saturated with particles has interesting implications for a microbial community response to a sudden influx of colloidal organic particles. Select cells that form surface particle films can function as nutrient hotspots for neighboring bacteria, where indirect or direct contact with particle-coated cells can potentially result in the efficient breakdown and nutrient uptake for contacting cells. For example, coupled interaction between particle-coated cells and surface-enzyme expressing cells could be a potential mutually beneficial strategy within certain microbial consortia. Further work is needed to determine the properties of such surface particle films on bacterial surfaces and the influence of their variability on the interactions between bacterial species.

## Surface attachment and capture of particles

Introducing colloids within circulating media and removing weakly-associated particles through fluid shear, we observed discrete patches or clusters of organic colloidal particles distributed on the bacterial surface. AFM cell surface imaging of isolates (*Flavobacterium* sp. BBFL7, *Pseudoalteromonas* sp. TW7, and Alteromonas sp. ALTSIO) in circulating FASW media with ribosome amendments show patches or clusters of particles as a common surface feature after exposure to ribosomes as extracellular organic colloids (Fig 1). Small individual patches or clusters of particles that are formed and retained upon few bacterial surfaces after an exposure period of approximately 60 mins to organic colloidal particles. Among the various cells, a few select cells were more populated with particulate features compared to many immediate neighboring cells that showed bare, particle-devoid surfaces or few individual particles. Cells within populations of the same isolated presented variable levels of particle attachment. Most cells within a population had significantly lower particle attachment or fewer intact particles retained upon the bacterial surface after interaction. AFM micrographs of select surface

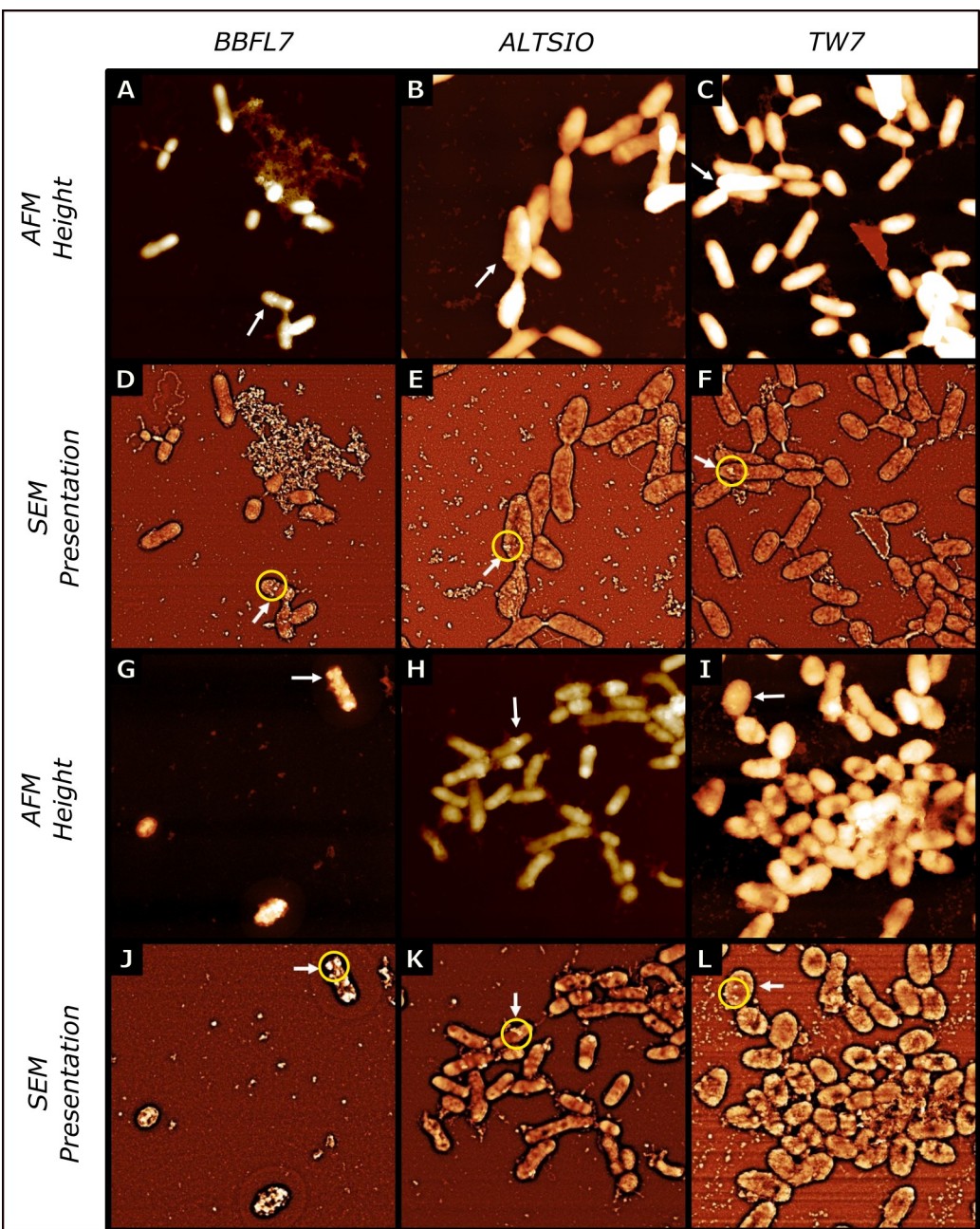

**Fig 1. Particle clusters and patches on bacteria surfaces.** AFM height images and respective SEM presentations of bacterial isolate surfaces during longer amendment with circulating ribosomes. *Flavobacterium* sp. BBFL7 (**A,D,G,J**), *Alteromonas* sp. ALTSIO (**B,E,H,K**), and *Pseudoalteromonas* sp. TW7 (**C,F,I,L**). The images show prominent surface patches of organic particles (white arrows) on select cells after exposure to ribosomes for 1 h (**A-F**) and for 4.5 h (**G-L**). Panel image scan sizes are 10 μm × 10 μm.

particle clusters and associated line section profiles observed in ALTSIO and TW7 cells showed observed particle clusters had varying physical dimensions and ranged between 40–200 nm in width and between 10–30 nm in height (Fig 2). After longer exposure of 4.5 h, the surface patches become more numerous and larger contiguous regions (S3 Fig). The aggregation of organic colloidal particles on bacterial surfaces is heterogeneous and becomes more

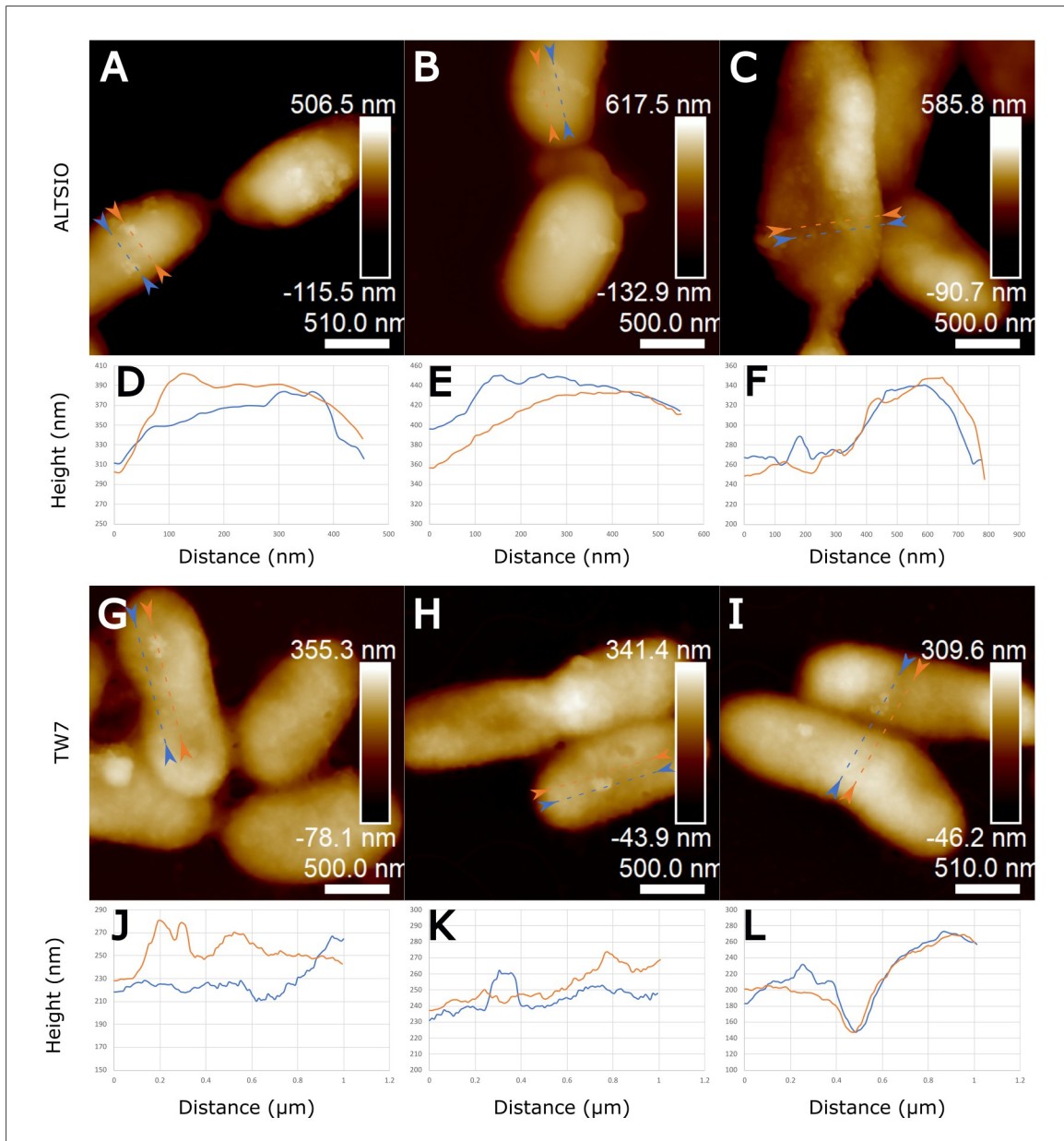

**Fig 2. Surface profiles of individual ribosome particle clusters.** AFM height images with demarcated line sections (blue and orange dashed lines and arrowheads) of surface-attached particle clusters of ribosomes on ALTSIO cells (**A-C**) and TW7 cells (**G-I**). Respective surface profiles of paired line traces show the height dimensions of such features with reference feature-free profiles from proximate cell surface regions (**D-F**: ALTSIO and **J-L**: TW7); corresponding to respective line sections in AFM images. Most discrete surface-attached particle clusters are roughly 40–200 nm in width and 10–30 nm in height, corresponding to several multiples of ribosome particles. Image scale bars provided.

unequal and extreme with a longer duration of colloidal particle exposure. More cells were observed with surface-associated particles, with more particle clusters observed in the interstitial spaces between individual cells. One notable observation is the expected increase of attached particles with increased exposure time, which contributes to developing more prominent surface particle patches. Ribosomes stick onto bacterial surfaces, forming protruding features shown in 3D representations of surface particle clusters and patches on individual

ALTSIO cells (S4 Fig). The frequency of particle attachment was variable for populations of cells within observed images. The formation of such prominent features may promote attachment of extracellular polymers or bacteria in potential biofilm formation, forming sites of physical intercellular interaction.

The observation of discrete surface clusters and patches in AFM imaging within circulating media, suggest intimate physical interaction of particles upon bacterial surfaces. Exposure of bacterial surfaces to elevated ribosomal particle concentrations in the ambient microenvironment leads to the development of discrete patches of particles distributed on the surface, as shown in Fig 1. Select cells appear to be preferentially disposed towards capturing or interacting with particles, even in a local neighborhood of cells. The retention of small individual patches or clusters of particles, formed upon few bacterial surfaces, after an exposure period of approximately 60 mins suggests the potential for intimate attachment to organic colloidal particles. With extended exposure time to ambient organic colloids, the proportion of cells with surface attached particles increases, and select cells present a disproportionately greater quantity of surface particle clusters. After longer exposure of 4.5 h, the surface patches become more numerous and contiguous as the bacterial surfaces become conditioned with attachment of organic colloids. The attachment of ambient particles onto preexisting patches can consequently contribute to the attachment of cells onto larger particle aggregates and other cells via surface patches [27]. We consider the differences in the development of particle films compared to surface particle patches likely due to differences in particle interaction frequency for suspended cells compared to surface-attached bacteria. Suspended cells are likely to have more frequent encounters with particles and develop substantial films which may obscure more intimately associated surface clusters or patches of particles. The particles patches can offer some insight into how various organic colloids interact with and condition bacterial surfaces. Significant variability was observed within cell populations of individual isolates, but different marine bacterial species may potentially vary in their interaction with colloidal particles. Physical properties, such as charge distribution, hydrophobicity and surface roughness can influence the interaction forces of individual particles with bacterial surfaces [45–48]. The spatial distribution of charged residues and hydrophobic structures on bacterial surfaces may contribute to the attractive forces promoting the retention and localized clustering of surface particles in specific regions.

We consider the distribution of surface-attached particles on bacterial surfaces to be random, but more work is needed to consider the possibility of site-specific particle attachment. Possible developments of surface-attached particles include the successive attachment of particles to form nucleation sites for particle clusters or attachment sites for other bacteria. It is possible that organic colloids preferentially attached onto each other compared to bacterial surface, and the surface capture of particles can result in effective surface-localized aggregation of particles. For instance, given a bacterial cell expressing clusters of surface particles, other bacteria can potentially interact with the cell and attach onto the surface around the particle cluster, limiting the desorption and loss of particles, allowing both cells to benefit from degradation and nutrient uptake of the associated particles.

The presence of particle patches on the bacterial surfaces has potential implications in microbe–microbe interactions. Particle clusters on cell surfaces can become nucleation sites for aggregation and association with other labile colloidal DOM and potentially refractory DOM. These features can also become sites bacterial cell attachment to other surfaces, functioning as patchy sites of increased surface adherence [27]. Another possibility is that particle clusters can be surrounded by bacterial surface enzymes, resulting in the breaking down of particles within an enclosed pocket. For example, bacterial cells attached together via particle clusters can retain organic particles on their surfaces after dissociation and disruption of

particle clusters. The aggregation and distribution of organic colloidal particles through contact-dependent cell interactions could result in physical associations where some bacterial species or subpopulations can derive benefits from scavenging colloidal DOM from other cell surfaces. This potential mechanism is likely driven due to random encounters, and more investigative work is needed to determine if specific mechanisms exist to express such strategies to capitalize on elevated levels of ambient organic colloids.

## Observed surface features relevant for bacterial-surface particle attachment

Surface-attached bacteria were observed with altered surface features (e.g., surface corrugations, surface pits) that can influence their interactions with organic colloidal particles like ribosome particles. Compared to discrete surface-attached particle clusters of ribosomes, some TW7 bacteria cell surfaces were observed to become corrugated with nanoscale patches of the surface raised in a semiregular pattern. AFM data shows that the corrugation patterns were similar among cells within an image field of 10 μm × 10 μm (Fig 3A–3D). Line section surface profiles of select corrugation features indicate such features are approximately 20 nm in height and range between 50–100 nm in width (Fig 3E). The corrugation features contribute to rough surfaces with colloid size pocket regions on bacterial surfaces. In addition to surface corrugation, certain TW7 bacterial surfaces were observed with prominent membrane pit features. TW7 Bacterial cells were observed with pits on the surface ranging from 50–300 nm in width and 10–50 nm in depth, with variable surface pit sizes and depths, as shown with respective line traces shown in Fig 4A. Surface pits were observed during live cell imaging of TW7 cells of an intact cell over a period of 92 minutes (Fig 4B). Data from consecutive horizontal line trace profiles across the cell are summarily plotted at 0-, 45- and 92-minute time points. The time-lapse AFM showed a consistent expression and physical structure of the surface pit feature measuring approximately 50 nm deep and 300 nm wide throughout the duration of imaging. Observed surface pits in TW7 cells bear resemblance in size and structure to mouth-like pit structures observed in certain soil-isolated *Sphingomonas* bacteria [49]. Surface pits in *Sphingomonas* A1 strain are 0.02–0.1 μm in diameter and part of the pleat-like surface structure involved in breakdown and uptake of intact alginate particles [50].

Marine bacteria express potential adaptive surface features that have potential implications in modulating surface interactions with ambient organic colloidal particles. One observed adaption was the development of colloid-sized surface corrugations and raised protrusions modifying the bacterial surface. These raised features increase the surface area available for particle attachment, where select areas between features have increased contact area for particle attachment. Corrugation features are predicted to reduce surface energy barriers for colloidal and contact interactions, influencing the attachment rates of ambient particles onto bacterial surfaces [51]. The variations in the potential particle contact area can form different areas for stronger and weaker attachment of organic particles on the bacterial surface. Such features may imply surface remodeling of the bacterial outer membranes similar to membrane vesicle production, with potential changes in of the distribution of surface and periplasmic proteins [52]. The observed corrugated features may result in the increased surface concentration of organic matter that is accessible to surface hydrolytic enzymes that produce soluble nutrients for uptake [13, 53]. This mechanism can influence bacterial responses in degrading colloidal particles and variable DOM adsorption during exposure to pulses of organic matter [54, 55]. The size similarities between the raised corrugation features and organic particles raises potential possibilities in their involvement in surface interactions with particles.

Bacterial surface pits were another type of observed adaptive bacterial surface features, where cells can form discrete membrane pockets and depressions for contained interaction

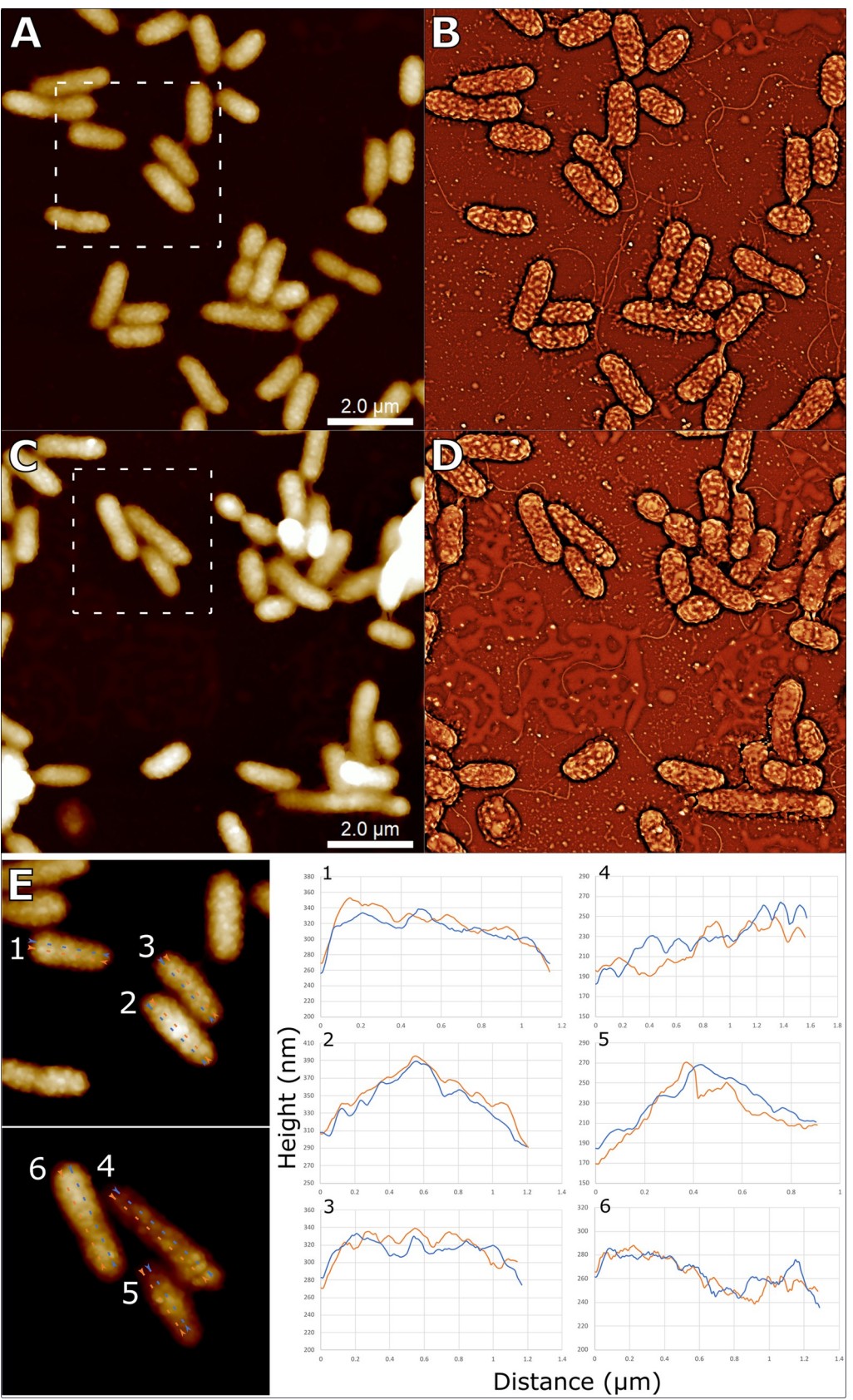

**Fig 3. Bacterial surface corrugation.** AFM micrographs *(A, C)* and representative SEM presentations *(B, D)* of TW7 cells expressing surface corrugations. *(E)* Magnified images of cells with marked line section crossing the surface corrugation features with respective surface height profiles *(1–6)*. Image scale bars provided.

sites with organic colloidal particles and DOM. Based on the breadth and depth of surface pits, such features have the potential to contain small aggregates of organic colloidal particles with a contained space, with a potential artificially elevated local concentration of organic matter. Such features may have implications for the possibility of the degradation and direct uptake of organic colloids [56]. The expression of superchannels involved in the direct uptake of alginate have been observed previously, and such features may be involved in similar mechanisms [49]. One potential feature of surface pits is the possible formation compartmentalized enclosure around particles between interaction cells, where breakdown of nutrients if directly available to benefit the associated cells. The expression of surface pits has greater implications for bioremediation in bacteria-directed breakdown and uptake of particles and pollutants with a combination of surface pits and superchannel structures [57].

The observations of the corrugated bacterial surface expression and pit formation were rare, less than < 1% of observed cells but these processes are likely to be generally overlooked among marine bacterial assemblages in the surface waters. These observations point towards potential evidence of bacterial strategies in exploiting surface interactions with ambient organic colloidal particles. The corrugation adaption of bacterial surfaces increases the available surface area for interaction, potentially creating preferential attachment sites for extracellular particles. This process can facilitate interaction with other cells and ambient biopolymers and indirectly influence interactions with extracellular particles and induce the utilization of ribosome particles and other organic colloids. Our observations from TW7 cells suggest surface pit formation can be strategy used in rare instances by bacteria. Some taxa of marine bacterial may potentially express surface pits more frequently compared to other cells in an assemblage. Such bacteria can potentially use surface pits to process intact colloidal particles within a contained space, taking advantage of the confinement to generate a limited nutrient hotspot of organic matter localized around the cell. Further investigation is required to determine the conditions and mechanisms involved in the expression of these features and the consequential changes in the bacterial ecophysiology. Similarly, further research is needed to determine the mechanisms involved in surface pit expression and the variability among marine bacterial species. The presence of these adaptations has implications for variations in bacterial interactions and strategies of nutrient acquisition within marine microenvironments enriched with organic colloidal particles.

## Growth on colloids—Common among multiple isolates

The primary finding is the evidence of bacterial responses to the organic matter of colloidal ribosome particles through contact-dependent interactions. In growth response experiments, abundances significantly increased for all tested bacteria over a period of 18 hours when cultures were amended with ribosomes (3.3 $\mu g$ $mL^{-1}$) in comparison to the non-amended controls, as summarized in Table 1. Natural bacterial assemblages in whole-seawater (WSW) and 0.6-$\mu m$-filtrate increased ~2-fold, while most of the isolate's (ALTSIO, Tw7, Tw2, and SWAT-3) abundances increased ~10-fold. *V. cholerae* abundance also substantially increased (~5-fold). Relative growth yields for different bacteria suggest that some isolates are better than others at incorporating ribosomal carbon into bacterial biomass. The increase in bacterial abundance in combination with the depletion of ribosome particles in ALTSIO and 0.6-$\mu m$-

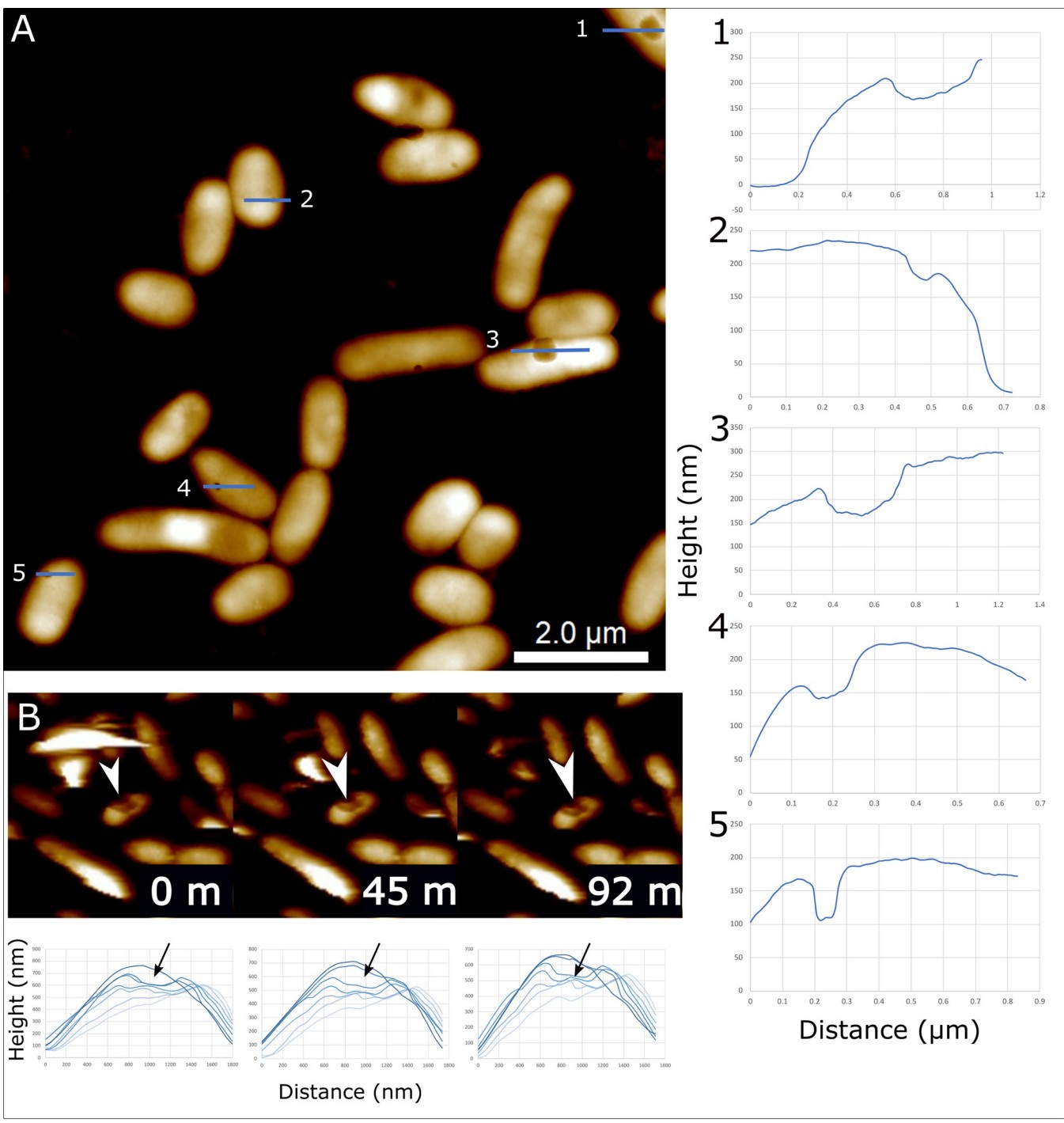

**Fig 4. Surface pits form in subset of bacteria and are stable structures in live cells.** *(A)* In TW7 cells, surface pits appear as depressions with variable cross-sectional profiles. Line section graphs *(1–5)* show the cross-sectional profiles of line segments labelled within the AFM image. *(B)* A surface pit (white arrow) was observed in fluid imaging of live TW7 cells over a period of 92 minutes, showing consistent pit dimensions and sizes, as shown with respective line traces. Image scan frames are shown at 0-, 45- and 92-minute time points with a scan area of 6.5 μm x 6.5 μm. Topographic data from consecutive horizontal line trace profiles across the cell are summarly plotted for each image, showing the height depression of the surface pit (black arrow).

**Table 1. Marine bacterial growth response to 18 hr incubation with ribosome amendment.**

| Isolate | Ribosome | Buffer | Control |
|---|---|---|---|
| WSW[a] | $1.03 \times 10^7 \pm 1.22 \times 10^6$ | $2.75 \times 10^6 \pm 5.30 \times 10^4$ | $2.63 \times 10^6 \pm 2.04 \times 10^5$ |
| 0.6-μm Filtrate[a] | $5.12 \times 10^6 \pm 8.70 \times 10^5$ | $1.46 \times 10^6 \pm 3.41 \times 10^4$ | $1.33 \times 10^6 \pm 1.32 \times 10^5$ |
| ALTSIO[c] | $2.58 \times 10^7 \pm 2.08 \times 10^6$ | $2.15 \times 10^6 \pm 4.92 \times 10^4$ | $2.52 \times 10^6 \pm 9.69 \times 10^4$ |
| TW7[c] | $1.30 \times 10^7 \pm 1.12 \times 10^5$ | $1.13 \times 10^6 \pm 2.51 \times 10^4$ | $9.68 \times 10^5 \pm 5.11 \times 10^4$ |
| TW2[c] | $1.83 \times 10^7 \pm 2.66 \times 10^5$ | $1.36 \times 10^6 \pm 3.11 \times 10^4$ | $1.00 \times 10^6 \pm 2.08 \times 10^4$ |
| SWAT-3[c] | $4.82 \times 10^6 \pm 7.04 \times 10^5$ | $3.26 \times 10^5 \pm 2.25 \times 10^4$ | $1.13 \times 10^5 \pm 5.77 \times 10^4$ |
| BBFL7[c] | $1.88 \times 10^6 \pm 2.95 \times 10^6$ | $2.09 \times 10^5 \pm 1.57 \times 10^4$ | $2.06 \times 10^5 \pm 5.43 \times 10^4$ |
| V.C. [c] | $3.86 \times 10^6 \pm 7.97 \times 10^5$ | $6.52 \times 10^5 \pm 1.49 \times 10^5$ | $6.54 \times 10^5 \pm 2.60 \times 10^5$ |

Bacterial inoculum abundance ± SD (cells mL$^{-1}$): (a) $3.5 \times 10^5$, (b) $2.8 \times 10^5$, (c), $5.0 \times 10^4$

filtrate support the hypothesis that marine bacteria can readily degrade and utilize organic ribosomes in the ocean.

Bacterial growth is observed for different marine isolates and assemblages at an amendment concentration of 3.3 μg mL$^{-1}$ suggesting sufficient association, degradation, and utilization of ribosomes to support bacterial cell production that can involve direct contact and association. The addition of ribosome particles results in the coating of bacterial surfaces with a film of contiguous particles, effectively altering their surface properties and surface-mediated interactions with the marine microenvironment [55]. The utilization of specific types of organic colloidal particles raises the possibility for potential bacterial adaptive strategies in transforming particle and DOM distribution at a submicron scale in the marine microenvironment. Various bacterial species can have varying potential for interaction and thrive on enrichments of colloidal DOM [21].

Apparent bacterial consumption for ribosome particles is indicative of the utility of other similarly-sized organic particles as a labile organic source of carbon and nitrogen for bacterial production. The breakdown of colloidal particles is an adaptive strategy with inherent limitations in the advantage and utility of organic colloids for bacterial utilization. Fragmented particles and soluble products that remain suspended and unattached may be lost to bacteria due to the transience of the interaction. Bacterial surfaces can function to generate nanoscale hotspots as contiguous films of particles form due to particle attachment at high ambient concentrations of colloidal organic particles. Such mechanisms potentially influence cooperative strategies through self-organization to limit diffusive loss of substrate particles [58]. This transformation of organic matter has implications for the residential times of nascent DOM in the surface waters and its vertical transport in the water column. From the observed utilization of amended ribosomes, it can be hypothesized that different isolates have different potential in processing them, where select members in a bacterial assemblage can derive a disproportionate advantage from influxes of organic colloidal particles [21]. Further investigation is needed to quantify the variability of such utilization and the bacterial respiration and growth efficiency rates with respect to ambient concentrations of specific organic colloids, and the unknown factors that influence these rates.

Overall, the presence of extracellular colloids has many possibilities for influencing individual microbes. In elevated conditions of colloidal concentrations in local microenvironments, 3.3 μg mL$^{-1}$ of ribosomes (~$10^{11}$ ribosomes mL$^{-1}$), most marine bacteria in our study were responsive and utilized the colloidal DOM nutrient amendments to increase bacterial abundance. The initial immediate response is in the surface being covered with ribosomal particles, occasionally resulting in the formation of contiguous films, due to the high concentration of

particles. The surface adsorption of particles alters properties of bacteria, influencing the biological properties of surface proteins [55]. The observation of bacterial surfaces saturated with particle films has ecological implications for functional partitioning of microbial communities, wherein specific bacterial taxa can incorporate colloidal organic particles [59, 60]. The observation of surface particle films can correlate functional specificity of particular colloidal particles with nanoscale contact-dependent features for the efficient breakdown and nutrient uptake for interacting bacterial cells. Intimate physical interactions between particle-associated cells with neighboring cells may be a potential mutually beneficial strategy within bacterial consortia that can couple bacterial diversity with the chemodiveristy of organic colloidal particles [21]. Further work is needed to determine the properties of such surface particle films on bacterial surfaces and the influence of their variability on the interactions between bacterial species.

## Ecological implications of bacterial particle capture

The ability for an individual heterotrophic bacterium to preferentially capture organic colloidal particle has ecological importance for its fate and survival within a bacterial community. Such bacteria can potentially become a transient nutrient hotspot by hosting discrete aggregates of labile organic colloids directly on their surface to their benefit or expense. The secure attachment of organic colloidal particles forms small clusters and patches of particles, which in excess can smother the surface proteins involved in organic matter degradation and nutrient uptake. A behavior that can develop nanoscale nutrient sources to influence bacterial physiology and behavior has the corresponding risk of inundating the surface with particles and blocking surface biological activity. Bacterial surface adaptions, such as surface corrugations and pits, can potentially modulate particle attachment for cells to take advantage of conditions of elevated organic particles. As potential intermediaries, organic particles can influence bacterial interactions within an assemblage as contact-dependent strategies for nutrient acquisition. A potential niche for particle-aggregating bacteria has the benefit of deriving physiological benefits from the enzymatic activity of neighboring bacteria. As such, directed bacterial surface interactions with organic colloidal particles can promote various bacterial associations and interactions with their microenvironment.

## Conclusion

This study demonstrates that marine bacteria tested have the capacity for direct particle attachment onto their surfaces, as contiguous particle films and discrete nanoscale particle clusters and patches. Longer exposure to elevated ambient colloidal concentrations yields higher frequency and greater size of particle patches as more particles and smaller patches coalesce together. Select cells within bacterial population can promote cell interactions by taking advantage of particle clusters using adaptive features such as surface corrugations and surface pit formation. The presence of these particle patches has implications for the influence of colloidal fraction of DOM on microbe-microbe interactions and microbial community structuring. Furthermore, the discovery of surface pits in marine bacteria suggests the potential involvement of superchannel systems for the degradation and uptake of colloidal organic matter. The biological interactions with specific forms of colloidal organic matter can influence their residence time and lability of organic colloids. Further investigation is needed to determine the influence of various bacterial taxa on the interactions and cycling of colloidal organic matter. Understanding such behaviors can further the understanding of how marine bacterial communities can adapt and respond to discrete influxes of colloidal DOM into marine microenvironments from various phenomena such as in mass cell lysis events.

## Materials and methods

### Bacterial growth response

Seawater was collected from the Ellen Browning Scripps Memorial Pier (San Diego, CA, US) for preparation of three experimental conditions: unprocessed whole seawater (WSW), 0.6-μm filtered seawater (0.6-μm filtrate) and filtered autoclaved seawater (FASW). WSW and 0.6-μm Filtrate were diluted 10x with FASW at the start of the experiment to achieve initial bacterial cell densities of $3.5 \times 10^5$ and $2.8 \times 10^5$ cells mL$^{-1}$, respectively. Marine bacterial isolates *Alteromonas* sp. (ALTSIO), *Pseudoalteromonas* sp. (Tw7), *Pseudoalteromonas* sp. (Tw2), *Vibrio* sp. (SWAT-3), and *Vibrio cholerae* strain 2740–80 were grown with shaking in Zobell 2216E medium overnight, diluted 1: 1000 in filtered autoclaved seawater (FASW), and acclimated for 48 h at 23°C. Cultures were then diluted into fresh FASW to an initial cell density of $5 \times 10^4$ mL$^{-1}$ at the start of the experiment. Please see S1 Table for phylogenetic classification and source of all isolates used in this study.

Three experimental conditions (WSW, 0.6-μm filtrate, and isolate cultures) for three treatments were setup for each of the three experimental conditions. Specific treatments used in each condition included the following: 0.3 or 3.3 μg *E. coli* ribosomes (New England BioLabs Cat# P0763S, Ipswich, MA USA) mL$^{-1}$ culture (hereafter "R"); ribosome buffer (final concentration = 2 μM Hepes-KOH, pH 7.6; 1 μM Mg(OAc)$_2$; 3 μM KCl; 0.7 μM beta-mercaptoethanol) or no amendment. Experiments were conducted in triplicate, using culture volumes of 10 mL in 14 mL polycarbonate tubes, with rotary shaking, at 23°C.

Bacterial abundance for the five treatments was determined at 0 h and 24 h (n = 120). Samples for bacterial abundance were fixed with formalin at a final concentration of 2% and frozen at -80°C. Upon thawing, samples were immediately placed on ice and stained 1x with SYBR Green I (Invitrogen, Molecular Probes Cat# S7563, Carlsbad, CA USA). Bacterial cells were enumerated with a BD Accuri C6 Plus flow cytometer (Ex. 488nm, Em. 533). Experimental treatments for WSW, 0.6-μm filtrate and ALTSIO cultures were sampled at 0 h and 24 h, and fixed and filtered on 0.22-μm polycarbonate filters (GTTP, Millipore). Filters were affixed to a glass slide with adhesive tabs for atomic force microscopy imaging.

### Ribosome exposure experiments

AFM experiment was designed to expose select bacterial cultures to a continuous flow of media with suspended ribosomes to minimize the effect of potential colloid settling onto bacterial surfaces under otherwise relatively static conditions. Isolated cultures of *Alteromonas* sp. ALTSIO, *Pseudoalteromonas* sp. TW7, and *Flavobacterium* sp. BBFL7 cells were grown in Marine Broth 2216 liquid media. Overnight cell cultures were diluted tenfold into fresh marine broth medium and incubated for 6 hrs. 1 mL of culture was centrifuged at 3000 g for 5 minutes and resuspended with 0.02-μm syringe-filtered FASW (Whatman® Anotop®) to a final volume of 400 μL before application to AFM sample substrates. Substrates were prepared by securing freshly cleaved 12 mm mica discs onto clean glass slides using epoxy and treated with 1 mg mL$^{-1}$ poly-lysine solution to promote cell adhesion. 100 μL of the cell suspension was deposited onto mica and placed onto the AFM apparatus. The imaging fluid medium was circulated using syringe pumps (Harvard Apparatus) in withdrawal and infusion modes to continuously exchange fresh 0.02-μm Anodisc syringe-filtered FASW amended with ribosomes (conc. approx. 0.66 μg mL$^{-1}$ ($1.8 \times 10^{11}$ particles mL$^{-1}$)) at a rate of 10 μL min$^{-1}$. After 1 hr, samples were fixed with formalin (20 μL of 37% formalin) for 15 minutes and rinsed with 1 mL of HPLC grade water 5 times. Samples were dried in a laminar flow hood and stored in plastic petri dish until imaged with AFM.

## AFM imaging

AFM imaging was performed on a Dimension FastScan atomic force microscope (Bruker, Santa Barbara, CA USA) in PeakForce Tapping™ Mode using ScanAsyst-Air probes (nominal spring constant: 0.4 N/m and tip radius: 5 nm, Bruker) for fixed and air-dried samples. Sample imaged under fluid conditions were imaged in 0.22-μm-filtered autoclaved seawater with ScanAsyst-Fluid probes (nominal spring constant: 0.7 N/m and tip radius: 20 nm, Bruker). Acquired AFM micrographs were processed and analyzed using Nanoscope Analysis (Bruker) and Gwyddion software (http://gwyddion.net). Raw AFM height image data, generated from the feedback-controlled scanning tip motion, was processed using minimal line-flattening and plane-fitting routines. Peak force error image data, generated from the setpoint error of the applied peak forces by the scanning tip, were minimally processed using low-pass filters. Select AFM images were processed to improve visualization of surface features using Gwyddion software to generate SEM image presentations from height data simulated by Monte Carlo integration [61].

## Supporting information

**S1 Fig. Nonspecific binding of ribosomes to cells and particle films.** Fluorescence images show bacterial cells (red: FM 4-64fx) variably covered with ribosome (green: SYBR Green II), when amended to a ribosome concentration of $5 \times 10^9$ particles mL$^{-1}$ (before **(A)** and after **(B)** 60s ribosome amendment). AFM peak force error images of cells from 0.6-μm-filtrate **(C)** and a marine isolate Alteromonas sp. ALTSIO cell **(D)** are associated with small films or clusters of ribosome particles (dashed regions). Cells were observed after ribosome amendment 3.3 μg mL$^{-1}$ (= $8 \times 10^{11}$ particles mL$^{-1}$). **(E)** AFM peak force error image showing a cell from natural assemblage with the surface covered by large patches of particles (dashed regions) after amendment and exposure to ribosomes.
(TIF)

**S2 Fig. Cell depletion of available ribosomes.** AFM imaging study of bacterial depletion of added ribosomes. Ribosomes were amended to bacterial isolate ALTSIO and 0.6-μm-filtrate natural assemblage cells and incubated for 24 h. Results show the depletion of ribosomal particles (red dashed regions) for 0.6-μm seawater filtrate cells **(A, B)** and Alteromonas sp. ALTSIO cells **(C, D)**. Mean curvature images processed from topographic data show fewer ribosome particles (white particle features) on the background 0.22-μm polycarbonate filter at 0 h, minutes after amendment, **(A, C)** and after 24 h incubation **(B, D)**.
(TIF)

**S3 Fig. Changes in surface attached ribosome attachment after extended exposure to ribosomes.** AFM **(A, B)** and respective SEM presentations **(C, D)** of Alteromonas sp. ALTSIO cells. Many cells have substantial surface patches of ribosomes after extended amendment times (4.5 h) to high concentrations. Example surface patches are indicated by white arrows and white circles. In certain regions, surface patches appear to coalesce into larger patches, towards forming a contiguous film of particles that covers a group of cells. Panel image scan sizes are 10 μm × 10 μm.
(TIF)

**S4 Fig. 3D images of surface-attached particle clusters.** 3D surface representations of AFM topographic data **(A)** and respective SEM presentations **(B)** of an Alteromonas sp. ALTSIO cell with multiple surface particle clusters (white protruding features). **(C-F)** SEM presentation images showing different regions and features of ribosome attachment on bacterial surfaces,

where small groups of particles attach directly onto bacterial surfaces **(C, D)** or indirectly to cells **(E, F)**.
(TIF)

**S1 Table. Phylogenetic classification of marine bacterial isolates.**
(TIF)

## Author Contributions

**Conceptualization:** Nirav Patel, Ryan Guillemette, Farooq Azam.

**Formal analysis:** Nirav Patel, Ryan Guillemette.

**Funding acquisition:** Farooq Azam.

**Investigation:** Nirav Patel, Ryan Guillemette, Ratnesh Lal.

**Methodology:** Nirav Patel, Ryan Guillemette, Ratnesh Lal.

**Project administration:** Farooq Azam.

**Resources:** Farooq Azam.

**Supervision:** Ratnesh Lal, Farooq Azam.

**Visualization:** Nirav Patel.

**Writing – original draft:** Nirav Patel, Ryan Guillemette, Farooq Azam.

**Writing – review & editing:** Nirav Patel, Ratnesh Lal, Farooq Azam.

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
