## [Decision Letter · Decision Letter 0]

23 May 2022

PONE-D-22-10338Bacterial surface interactions with organic colloidal particles: nanoscale hotspots of organic matter in the ocean.PLOS ONE

Dear Dr. Patel,

Thank you for submitting your manuscript to PLOS ONE. After careful consideration, we feel that it has merit but does not fully meet PLOS ONE’s publication criteria as it currently stands. Therefore, we invite you to submit a revised version of the manuscript that addresses the points raised during the review process.

We look forward to receiving your revised manuscript.

Kind regards,

Wei-Chun Chin

Academic Editor

PLOS ONE

Journal Requirements:

3. Please upload a copy of Supporting Information Table 1 which you refer to in your text on page 24.

Additional Editor Comments:

please address questions and concern from two reviewers and revise the manuscript.

Reviewers' comments:

Reviewer's Responses to Questions

**Comments to the Author**

1. Is the manuscript technically sound, and do the data support the conclusions?

Reviewer #1: Yes

Reviewer #2: Yes

2. Has the statistical analysis been performed appropriately and rigorously? 

Reviewer #1: N/A

Reviewer #2: Yes

3. Have the authors made all data underlying the findings in their manuscript fully available?

Reviewer #1: Yes

Reviewer #2: Yes

4. Is the manuscript presented in an intelligible fashion and written in standard English?

Reviewer #1: Yes

Reviewer #2: Yes

5. Review Comments to the Author

Reviewer #1: This paper on ‘Bacterial surface interactions with organic colloidal particles: nanoscale hotspots of organic matter in the ocean’ is well written and articulated, and I recommend publication with minor changes.

General comments

For greater relevance it might be good to have a statement which compares conditions and concentrations in the experiments to those in natural seawater, e.g., the relative concentrations and ratio of ribosome colloids to bacteria.

Detailed comments

Line 522: ‘Isolate cultures’ should be ‘Isolated cultures’

Line

Line 676 and 679: What is ‘AFM error mode’?. Please explain

Line 683: ‘Ribosomes were amendment’ should be ‘Ribosomes were amended’

Line 689: What is ‘SEM-like presentations ’? Please explain

Line 697: What is ‘SEM simulated image’? Please explain

Reviewer #2: The manuscript demonstrated bacterial surface interactions with nano-sized colloidal particles including colloid-sized surface corrugations and protrusions. The data showed that ribosomes can attach onto bacterial surfaces and bacteria can readily utilize this organic matter as substrates to growth. This study provided useful information about the adaptive strategies of pelagic marine bacteria for colloid capture and utilization as nutrients. Overall, this study is interesting and easy to follow. Comments are listed in the bellow:

1. Line 33: the unit of organic colloid abundance should be “particles mL-1”

2. The manuscript lacks statistic data, only providing AFM image looks weak. Please added some statistic data or figure form height, thickness and size at different experiments and further compared them.

3. Did you consider measuring attached forces with AFM, the force measurements will help explain mechanisms of surface corrugations, attachment, and protrusions.

4. Ｍany references are missing in the manuscript, need citing references to support the discussion.

5. Line 283 which physical properties? Should further discuss

6. Line 443-450, need reference to support.

6. PLOS authors have the option to publish the peer review history of their article (what does this mean?). If published, this will include your full peer review and any attached files.

Reviewer #1: **Yes: **Peter H. Santschi

Reviewer #2: No

---

## [Author Response · Author response to Decision Letter 0]

14 Jul 2022

Review Comments to the Author:

Reviewer #1: This paper on ‘Bacterial surface interactions with organic colloidal particles: nanoscale hotspots of organic matter in the ocean’ is well written and articulated, and I recommend publication with minor changes.

General comments

For greater relevance it might be good to have a statement which compares conditions and concentrations in the experiments to those in natural seawater, e.g., the relative concentrations and ratio of ribosome colloids to bacteria.

Response: A statement has been included to compare the concentrations of ribosomes in seawater to the concentrations used in experiments as follows: “Extracellular ribosomes in ambient seawater, from 16S rRNA proxy measurements, are present at approximately 107 ribosomes mL-1 , and can reach elevated concentrations up to 109 ribosomes mL-1 in regions enriched after cell lysis” (line 114)

(NOTE: The line numbers referred to correspond to the “Revised Manuscript with Track Changes” file, not the unmarked-up file “Manuscript”.

Detailed comments

Line 522: ‘Isolate cultures’ should be ‘Isolated cultures’

Response: Corrected (line 548)

Line 676 and 679: What is ‘AFM error mode’?. Please explain

Response: The term “AFM error mode” was updated in the manuscript to “peak force error” for accuracy. A brief explanation of the “peak force error” imaging has been included (line 573). AFM error mode images are a fundamental part of atomic force microscopy where images are generated from the tracking error of tip movement. The text was updated to state that peak force error images are generated from setpoint error of applied peak forces during AFM images. 

Line 683: ‘Ribosomes were amendment’ should be ‘Ribosomes were amended’

Response: Corrected (line 752)

Line 689: What is ‘SEM-like presentations ’? Please explain

Line 697: What is ‘SEM simulated image’? Please explain

Response: We corrected the inconsistent terms of “SEM-like presentations” and “SEM simulated image” to “SEM presentations”. A brief statement was included to explain the processing of AFM height images into SEM image presentations of the topographic data for visualization purposes (line 575). 

Reviewer #2: The manuscript demonstrated bacterial surface interactions with nano-sized colloidal particles including colloid-sized surface corrugations and protrusions. The data showed that ribosomes can attach onto bacterial surfaces and bacteria can readily utilize this organic matter as substrates to growth. This study provided useful information about the adaptive strategies of pelagic marine bacteria for colloid capture and utilization as nutrients. Overall, this study is interesting and easy to follow. Comments are listed in the bellow:

1. Line 33: the unit of organic colloid abundance should be “particles mL-1”

Response: Corrected (line 36)

2. The manuscript lacks statistic data, only providing AFM image looks weak. Please added some statistic data or figure form height, thickness and size at different experiments and further compared them. 

Response: Our study resulted in qualitative data for describing the outcomes of bacterial surfaces exposed to colloidal particles. The collected data was primarily qualitative in denoting the presence/absence of surface features from attached particles. Quantitative description of these features was difficult in many cases, including cells with excessive particle attachment and in regions with touching or overlapping bacteria. In such cases, particles could not be adequately characterized compared to more prominent counterparts of one or two particles or clusters on an individual cell. This contributed to difficulties in characterizing particle attachment in an unbiased manner for statistical comparisons. Future studies could further explore statistical comparisons of features and observations based on the limited qualitative descriptions.

3. Did you consider measuring attached forces with AFM, the force measurements will help explain mechanisms of surface corrugations, attachment, and protrusions. 

Response: Our current study was primarily focused on nanoscale changes of bacterial surfaces after exposure to organic colloids. We did consider investigating the attachment forces of individual particles to collect data that may help explain mechanisms of particle attachment. Such studies will help explain mechanisms of initial attachment events of individual particles on bare bacteria and may be limited for surface clusters and aggregates as discussed in the manuscript. Such studies would involve applying AFM force spectroscopy to probe interaction forces of individual particles in constrained configurations and conditions and may not reflect the range of microenvironmental conditions experienced by individual bacteria interacting with colloidal particles.

4. Many references are missing in the manuscript, need citing references to support the discussion.

Response: References have been added to support various discussion statements (line 367-375, 431-438, 448-452)

5. Line 283 which physical properties? Should further discuss

Response: We have included a brief statement regarding the physical properties as follows: “Physical properties, such as charge distribution, hydrophobicity and surface roughness can influence the interaction forces of individual particles with bacterial surfaces [45-48]. The spatial distribution of charged residues and hydrophobic structures on bacterial surfaces may contribute to the attractive forces promoting the retention and localized clustering of surface particles in specific regions.” (line 289)

6. Line 443-450, need reference to support. 

Response: References have been added to support discussion statements (line 462-476)

---

## [Editor Report · Decision Letter 1]

19 Jul 2022

Bacterial surface interactions with organic colloidal particles: nanoscale hotspots of organic matter in the ocean.

PONE-D-22-10338R1

Dear Dr. Patel,

We’re pleased to inform you that your manuscript has been judged scientifically suitable for publication and will be formally accepted for publication once it meets all outstanding technical requirements.

Kind regards,

Wei-Chun Chin

Academic Editor

PLOS ONE
---

## [Editor Report · Acceptance letter]

16 Aug 2022

PONE-D-22-10338R1 

Bacterial surface interactions with organic colloidal particles: nanoscale hotspots of organic matter in the ocean 

Dear Dr. Patel:

I'm pleased to inform you that your manuscript has been deemed suitable for publication in PLOS ONE. Congratulations! Your manuscript is now with our production department. 

Kind regards, 

on behalf of

Dr. Wei-Chun Chin 

Academic Editor

PLOS ONE